# Noncanonical altPIDD1 protein: unveiling the true major translational output of the *PIDD1* gene

Frédérick Comtois[1], Jean-François Jacques[1], Lenna Métayer[1], Wend Yam DD Ouedraogo[2], Aïda Ouangraoua[2], Jean-Bernard Denault[3,4], Xavier Roucou[1,4]

Proteogenomics has enabled the detection of novel proteins encoded in noncanonical or alternative open reading frames (altORFs) in genes already coding a reference protein. Reanalysis of proteomic and ribo-seq data revealed that the p53-induced death domain-containing protein (or *PIDD1*) gene encodes a second 171 amino acid protein, altPIDD1, in addition to the known 910-amino acid-long PIDD1 protein. The two ORFs overlap almost completely, and the translation initiation site of altPIDD1 is located upstream of PIDD1. AltPIDD1 has more translational and protein level evidence than PIDD1 across various cell lines and tissues. In HEK293 cells, the altPIDD1 to PIDD1 ratio is 40 to 1, as measured with isotope-labeled (heavy) peptides and targeted proteomics. AltPIDD1 localizes to cytoskeletal structures labeled with phalloidin and interacts with cytoskeletal proteins. Unlike most noncanonical proteins, altPIDD1 is not evolutionarily young but emerged in placental mammals. Overall, we identify *PIDD1* as a dual-coding gene, with altPIDD1, not the annotated protein, being the primary product of translation.

## Introduction

In humans, protein-coding genes are typically annotated with a single functional ORF or coding sequence (CDS), which is typically the longest ORF. The corresponding transcripts contain the reference CDS or a variant of the CDS after alternative splicing, resulting in the expression of the reference protein (refProts) and of different isoforms. However, there is increasing evidence that some genes have more than one ORF and produce transcripts with more than one translated ORF (Chung et al, 2007). A consensus on the nomenclature of these additional noncanonical ORFs and their proteins has yet to be reached. The most commonly used terms are small ORFs and small ORFs-encoded proteins or peptides, microproteins, alternative ORFs and alternative proteins. In this study, we use the term "alternative ORFs and proteins" (altORFs and altProts) in contrast with reference ORFs and proteins (refORFs and refProts). In an initial study to measure the stoichiometry between two proteins endogenously expressed from the same human gene, the levels of the altProt altMID51 were found to be three to six times higher than those of the refProt in cell lines and tissues (Delcourt et al, 2018). AltMiD51 was later found to be part of the mitoribosome and has since been annotated by UniProt with the accession L0R8F8 (Brown et al, 2017; Rathore et al, 2018). The purpose of this example was not to suggest that altProts are generally expressed at a higher level than refProts, but rather to highlight what may be left out by ignoring them altogether.

The translation of proteins can be determined at the RNA level with ribosome profiling or ribo-seq, and at the protein level with MS-based proteomics. However, the lack of annotation of altORFs and altProts represents a significant challenge to their detection and the study of their functions. Indeed, the identification of peptides by comparing experimental MS/MS spectra against conventional databases results in the identification of refProts, and MS/MS spectra for peptides from altProts remain unmatched. To address this issue, several studies and proteogenomic resources combining ribo-seq and proteomics help annotate altORFs and altProts (Olexiouk et al, 2018; Li et al, 2021; Ouspenskaia et al, 2022; Manske et al, 2023; Sandmann et al, 2023). The proteogenomic resource OpenProt allows for the identification of more than one ORF per transcript and annotates all ORFs larger than 29 codons in the transcriptome of different species and their corresponding proteins (Leblanc et al, 2024). The reanalysis of large-scale ribo-seq and proteomics data with OpenProt allows for the retrieval of evidence of expression at the translatome (altORFs) and at the proteome (altProts) levels, respectively, and the identification of potential multicoding genes.

The P53-Induced Protein with a Death Domain gene (PIDD1) encodes a 910-amino-acid protein that is auto-catalytically cleaved at S446 into a 445-amino-acid N-terminal domain (PIDD-N) and a 465-amino acid C-terminal domain (PIDD-C). PIDD-C is further cleaved at S588 to produce PIDD-CC (Tinel et al, 2007). PIDD1 is

[1]Department of Biochemistry and Functional Genomics, Université de Sherbrooke, Sherbrooke, Canada   [2]Department of Informatics, Université de Sherbrooke, Sherbrooke, Canada   [3]Department of Pharmacology and Physiology, Université de Sherbrooke, Sherbrooke, Canada   [4]Centre de Recherche du Centre Hospitalier Universitaire de Sherbrooke (CRCHUS), Sherbrooke, Canada

Correspondence: Xavier.Roucou@USherbrooke.ca

capable of engaging with a variety of interactors and stimulating the DNA-damage response, centrosome surveillance, NFkB activation, and cell death (Weiler et al, 2022). Several mutations in PIDD1 have been linked to cerebral cortex malformations and intellectual disability, although the underlying molecular pathophysiology remains unclear (Sheikh et al, 2021; Zaki et al, 2021).

Here, we show that *PIDD1* is among the potential multicoding genes annotated by OpenProt with strong evidence of expression by ribo-seq and MS-based proteomics for a 171-amino acid-long alternative protein termed altPIDD1. AltPIDD1 and PIDD1 are co-expressed from two overlapping ORFs encoded by the same transcript. This previously unknown protein partially localizes in actin-rich cytoskeletal structures and is cleaved during apoptosis. In contrast to other noncanonical proteins which emerged in primates and are evolutionary young (Sandmann et al, 2023), Alt-PIDD1 emerged in placental mammals. The findings of this study highlight a further instance of a multicoding gene, thereby reinforcing the necessity of more detailed and accurate descriptions of the genome's coding potential.

## Results

### The human *PIDD1* gene encodes a second protein, altPIDD1, in addition to the conventional PIDD1 protein

*PIDD1* contains 16 exons, with the PIDD1 canonical coding sequence beginning in exon 2 and ending in exon 16 (Fig 1A). However, a second ORF spanning exons 2 and 3 with an initiation codon upstream of PIDD1 initiation codon, is annotated by OpenProt (Leblanc et al, 2024). This alternative ORF is predicted to encode a 171-amino acid protein termed altPIDD1 (Fig S1A). According to Ensembl, *PIDD1* produces two protein-coding transcripts, including the reference mRNA, i.e., the most conserved and the most highly expressed transcript (Table S1): ENST00000347755.10 and ENST00000411829.6 (Ensembl release 112, GRCh38.p14). No transcript encoding exclusively either of the ORFs are annotated. Consequently, if altPIDD1 is expressed, it indicates that PIDD1 and altPIDD1 are necessarily translated from the same mRNA.

The amino acid sequence of the two proteins is completely different because the altPIDD1 ORF is in the +3 reading frame relative to PIDD1 ORF in the +1 reading frame (Fig 1A). Thus, both proteins are coded in the same gene but are not protein isoforms. AltPIDD1 is particularly rich in proline residues and displays nine cysteines (Fig S1A). Several regions of the protein are predicted to be intrinsically disordered with protein binding activity (Fig S1B). The 3D structure predicted by AlphaFold2 indicates no regions of secondary structures (Fig S1C).

The reanalyzes of mass spectrometry (MS)-based proteomics raw data with OpenProt indicate that altPIDD1 is expressed in a variety of human cell lines and tissues (Table S2). Together with the reanalyzes of ribo-seq data, they also suggest that altPIDD1 is more expressed than PIDD1 (Fig 1B). The aggregation of ribosome profiles from all studies analyzed by GWIPS-viz (Michel et al, 2014) reveals that elongating (Fig 1C) and initiating (Fig 1D) ribosomes are significantly clustered on the altPIDD1 ORF and initiation codon,

respectively. Furthermore, the protein sequence coverage, i.e.,, the ratio between the total observed length of the protein sequence and the total length of the protein, is significantly higher for alt-PIDD1 than for PIDD1 (Fig 1E, Table S2). This high coverage was independently confirmed with PepQuery (Wen & Zhang, 2023), a targeted peptide search engine that enables identification of novel and known peptides in MS proteomics datasets (Fig 1E, Table S2). Furthermore, altPIDD1 and PIDD1 were detected in a variety of human cell lines and tissues (Table S2). For altPIDD1, ATPGHTGCLSP GCPDQPAR is the peptide detected with the highest number of confident MS/MS counts (Table S2). We also detected this peptide in a crude lysate of HEK293 cells and Fig 1F shows a representative example of a spectrum comparison with a synthetic peptide. The observed b- and y-ions suggest a similar fragmentation patterned to that of a synthetic peptide. In addition, the similarity based on spectral angle and Pearson's correlation coefficient provides compelling evidence for the accurate assignment of the spectrum.

The protein abundance of PIDD1 is in the lower quartile in humans, ranking 16,593 among the 19,338 proteins quantified (Wang et al, 2015). The ribo-seq and MS data above collectively suggest that altPIDD1 is the main translational product of *PIDD1* at both the translatome and proteome levels. To test this hypothesis, the levels of PIDD1 and altPIDD1 were measured in HEK293 cells using a multiple reaction monitoring approach based on AQUA peptides (Liebler & Zimmerman, 2013; Delcourt et al, 2018), with one proteotypic peptide for altPIDD1 (ATPGHTGCLSPGCPDQPAR) and two for PIDD1 (LQSLPASLAGLR and VNLIALQR). The absolute quantification indicated that altPIDD1 is the most abundant product of *PIDD1*, with a stoichiometry of 39.5 (Fig 1G). Overall, these data confirm that altPIDD1 is the primary protein product of *PIDD1* gene expression, and not the canonical PIDD1 protein in HEK293 cells.

### Validation of the co-expression of altPIDD1 and PIDD1 from a single transcript

To further characterize the translation of both ORFs from the same transcript, we expressed different constructs in HEK293 cells (Fig 2A). Monocistronic constructs altPIDD1[HA] and PIDD1[Flag] were used as controls to verify PIDD1 and altPIDD1 expression by immunoblotting using anti-HA and anti-Flag antibodies, respectively. Dual-coding constructs include the endogenous 5'UTR and were used to test the co-expression of both proteins from the same mRNA with the initiation codons for altPIDD1 and PIDD1 in the normal nucleotide context. It should be noted that the presence of the endogenous 5'UTR ensures that translation initiation is not artificially forced at the start of the altPIDD1 ORF, as would be the case if the construct started with the altPIDD1 ORF.

PIDD1 (~100 kD) undergoes constitutive auto-processing at residues S446 and S588 to generate two C-terminal fragments, PIDD-C (~51 kD) and PIDD-CC (~37 kD) (Tinel et al, 2007) (Fig 2B). In cells transfected with PIDD1Flag, we observed the typical multiple band pattern after auto-processing (Tinel et al, 2007). Full-length PIDD1 was not consistently observed in all experiments, suggesting that self-processing is efficient. In cells transfected with PIDD1[(HA) Flag], where (HA) indicates that the tag is silent within the reading frame of PIDD1 but not within the reading frame of altPIDD1, we

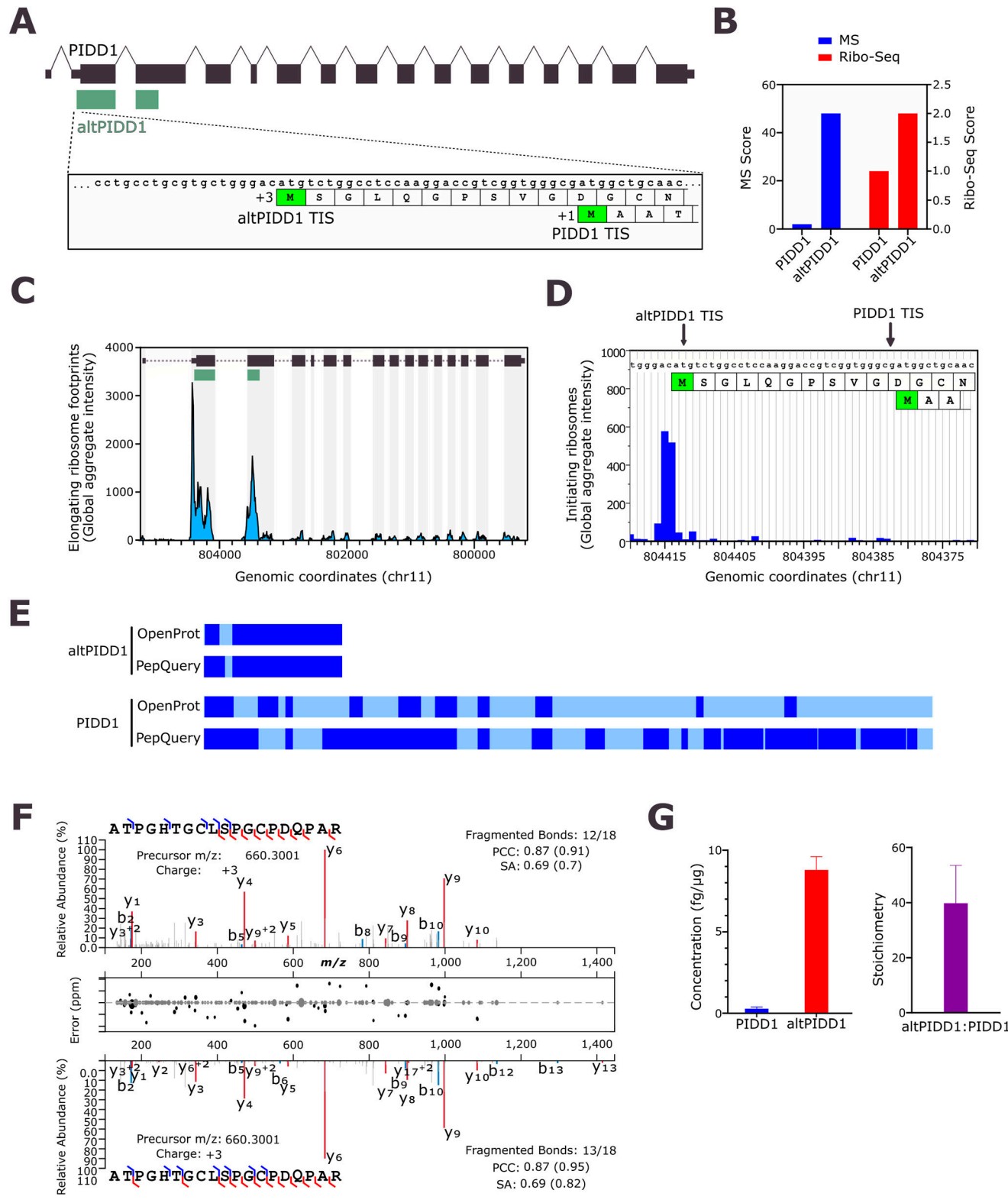

**Figure 1. *PIDD1* encodes two proteins and altPIDD1 is the main translational product, not PIDD1.**
**(A)** Schematic representation of dual-coding human PIDD1 variant 1 mRNA (RefSeq NM_145886, Ensembl ENST00000347755) containing 16 exons. Large boxes represent coding regions; thin boxes represent the regions annotated as untranslated (UTRs) in the mRNA. *PIDD1* ORF (black) is shared between exons 2–16 and altPIDD1 ORF (green) is shared between exons 2 and 3. Also shown are the nucleotides sequence around the translation initiation sites for PIDD1 and altPIDD1, and the first N-terminal residues. **(B)** MS and Ribo-Seq scores for altPIDD1 (OpenProt identifier IP_191523) and PIDD1 (UniProt identifier Q9HB75) extracted from OpenProt 1.6. **(C)** Sequence coverage for PIDD1 and altPIDD1 from the reanalysis of proteomics data with OpenProt 1.6 and PepQuery 2.0. Regions of the protein detected with unique peptides or undetected are

detected the co-expression of both PIDD1 and altPIDD1 (Fig 2C). Deletion of the predicted initiation codon for altPIDD1 in the PIDD1$^{\emptyset(HA)Flag}$ construct prevented the expression of altPIDD1 but not PIDD1, confirming the predicted translation initiation site for altPIDD1 (Fig 2D).

The data in Fig 1 show that altPIDD1 is expressed at higher levels compared with PIDD1. To compare the expression levels of the two proteins by Western blot directly, we inserted the same HA tag at their C-terminus and controlled expression using a doxycycline-inducible transcription activator (Tet-On system). At any concentration of doxycycline, altPIDD1 seems to be expressed at higher levels than PIDD1 and its processed forms (Fig 2E). Although altPIDD1 and the self-processed fragments of PIDD1 do not have the same molecular weight and therefore probably do not transfer with the same efficiency onto the immunoblotting membrane, the strong difference in expression level observed is certainly significant. Strikingly, altPIDD1, but not PIDD1, could be detected in the absence or at very low concentrations of doxycycline (i.e., 0–0.05 mg/ml), indicating that the preferential expression of altPIDD1 from the dual-coding mRNA becomes exclusive at low mRNA levels. Expression in the absence of doxycycline likely results from background leakage of the Tet-On system.

The elevated expression level of altPIDD1 relative to PIDD1 may be attributed to enhanced protein stability, elevated translation by ribosomes, or a combination of these mechanisms. To investigate the stability of PIDD1 and altPIDD1, we conducted a cycloheximide assay (Fig 2F). In the presence of this inhibitor of protein synthesis, PIDD1 undergoes fast auto-processing compared with PIDD1-C, resulting in the accumulation of PIDD1-CC whose level increases over time. AltPIDD1 exhibits relatively stable levels over the course of the experiment. In a control experiment, cycloheximide treatment resulted in a decreased level of the MYC protein. Overall, both proteins demonstrate high stability, and the high ratio of altPIDD1 relative to PIDD1 (Fig 1) is likely the consequence of enhanced translation of altPIDD1 by ribosomes, in agreement with the ribo-seq data (Fig 1B–D).

The optimal consensus sequence for the translation initiation site in mammals is the Kozak motif GCCRCCAUGG (where R at −3 is A or G; the AUG is the initiation codon). The A or G at position −3 and G at position +4 are important nucleotides for high translational levels (Kozak, 1986). Although a G is present at −3 from altPIDD1 initiation codon, the nucleotide at position +4 is a T rather than the key nucleotide G (Fig 2G, left panel). Because the initiation codon of altPIDD1 does not have an optimal Kozak motif and is located upstream of that of PIDD1, it is likely that translation of PIDD1 occurs via a leaky scanning mechanism (Kozak, 2002), after ribosomes ignore the altPIDD1 initiation codon and instead continue to the next initiation codon, that of PIDD1. This hypothesis can be tested by optimizing the altPIDD1 initiation codon to match an optimal Kozak context because of this modification should increase recognition of

the altPIDD1 initiation codon and decrease leaky scanning (Hernández et al, 2019). Indeed, a significant decrease in PIDD1 expression was observed after changing the nucleotide context of altPIDD1 translation initiation site to match the Kozak motif (Fig 2G, right panel) confirming the leaking scanning hypothesis in three different cell lines.

### AltPIDD1 partially localizes to actin-rich structures and interacts with proteins from the actin network

We examined the subcellular localization of altPIDD1 by immunofluorescence in HeLa cells. Both proteins localized to the cytoplasm whether co-expressed from the dual-coding PIDD1$^{(HA)Flag}$ construct (Fig 3A) or independently from PIDD1$^{Flag}$ and altPIDD1$^{HA}$ constructs (Fig 3B). The cytoplasmic localization of PIDD1 confirms previous observations (Janssens et al, 2005; Tinel et al, 2007). AltPIDD1 also appears to accumulate in discrete regions at the cell periphery and in cytoplasmic filaments (Fig 3B, arrows), suggesting that it may localize to cytoskeletal structures such as filamentous actin (F-actin). To test this hypothesis, we used a combination of physical and genetic strategies. First, we performed scratch wound assays in HeLa cells to induce the actin-based protrusions lamellipodia and contractile stress fibers. AltPIDD1 strongly accumulated in stress fibers (Fig 3C, left panel) and in the lamellipodial protrusions (Fig 3C, right panel). Second, we tested the effect of small GTPases that regulate actin dynamics; constitutively active RhoA-Q63L and Rac1-Q61L induce the formation of stress fibers and lamellipodia, respectively (Subauste et al, 2000). AltPIDD1 accumulated in stress fibers induced by RhoA-Q63L (Fig 3D, left panel) and in lamellipodia induced by Rac1-Q61L (Fig 3D, right panel). In these experiments, a minimum of 100 cells were observed for each replicate and altPIDD1 always localized in actin structures labeled with phalloidin. Taken together, these observations suggest that altPIDD1 interacts with actin-containing cytoskeletal structures.

To identify which proteins could interact with altPIDD1 in cells expressing altPIDD1$^{FLAG}$, we performed an analysis using Flag affinity purification and mass spectrometry (AP-MS) and used a no-bait control for quantitative comparison. A total of 173 interacting proteins were identified, with a minimum of two peptides and a SAINT score greater than 0.8. This represents a minimum of 2.81-fold enrichment compared with the control (Fig 4A). A gene ontology enrichment analysis to the human proteome on cellular component showed that several of the identified interactors are associated with different components of the cytoskeleton (Fig 4B), in agreement with the observed localization of altPIDD1. CAPN2 and CAPNS1, respectively, the catalytic and regulatory subunits of calpain-2, are present in the interactome (Table S3). CAPN2 was particularly enriched by AP-MS (Fig 4B, Table S3), and we validated this interaction in cells expressing altPIDD1$^{HA}$. Immunoprecipitation of endogenous CAPN2 pulled down altPIDD1$^{HA}$ (Fig 4C, left panel). In

shown with dark blue and light blue, respectively. **(D, E)** Public RiboSeq data of elongating (D) and initiating (E) ribosomes for PIDD1 were retrieved from GWIPS-viz. **(F)** Mirrored fragmentation spectra showing the acquired parallel reaction monitoring MS/MS spectrum from a unique endogenous (top) and synthetic (bottom) peptide from altPIDD1. b- and y-ions fragments are highlighted in blue and red, respectively. The Pearson correlation coefficient (PCC) and normalized spectral contrast angle (SA) are indicated. A peak-assignment/intensity difference plot is shown in the middle. **(G)** Absolute quantification in fg of protein per µg of whole cell lysate (left) and stoichiometry (right) of endogenous PIDD1 and altPIDD1 in HEK293 cells. Error bars represent SDs (biological triplicates).

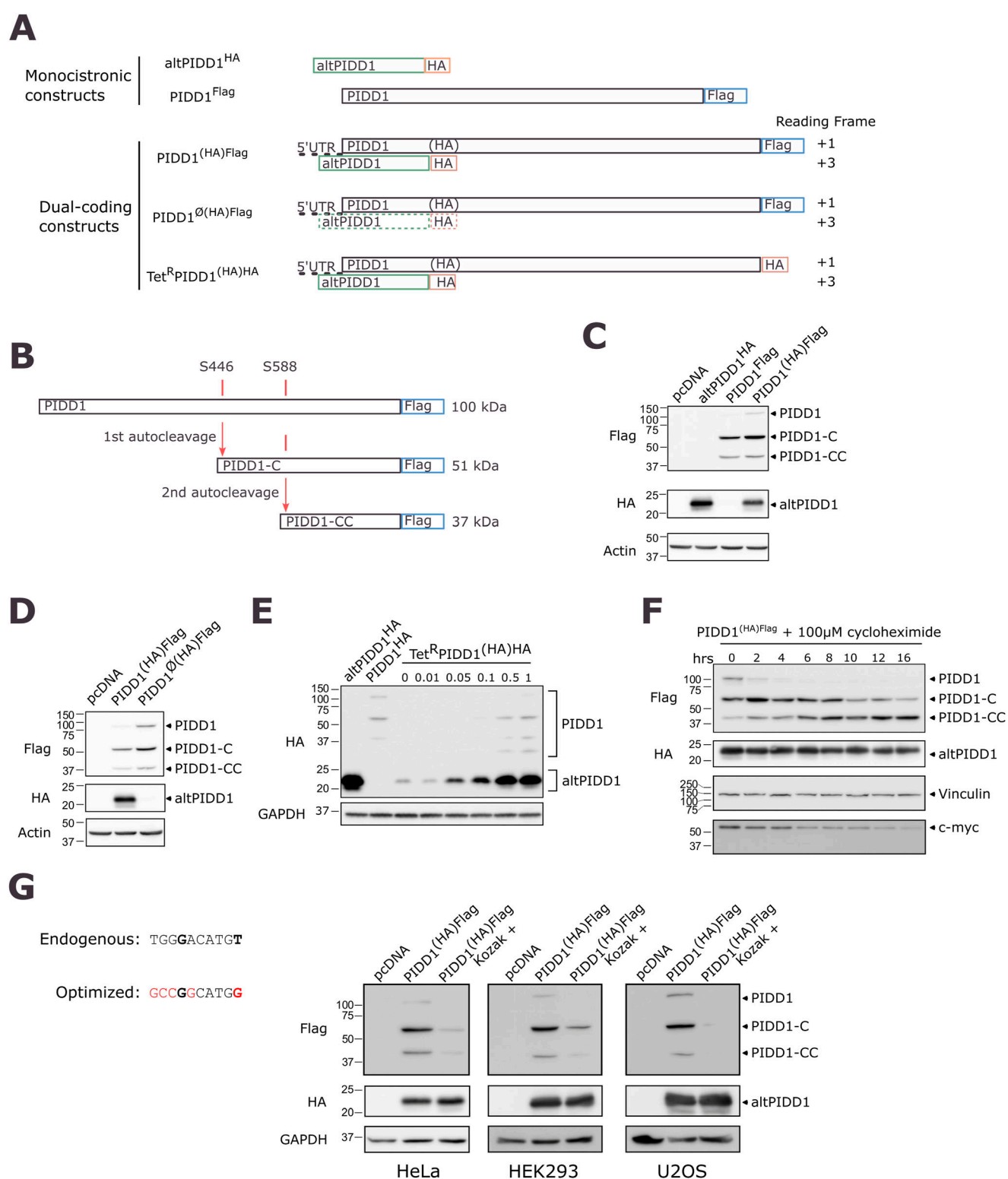

**Figure 2. Co-expression of PIDD1 and altPIDD1 from the same transcript.**
**(A)** Schematic representation of different constructs used to detect the expression and co-expression of altPIDD1 and PIDD1 by introducing an HA or a Flag tag at the C-terminus of each protein. For constructs encoding only PIDD1 or altPIDD1, the corresponding coding sequences were cloned without the 5′UTR context. Dual-coding constructs contain the endogenous 5′UTR to preserve the altPIDD1 initiation codon in its original/genomic context. For dual-coding constructs, the reading frame is indicated on the right; PIDD1 reading frame is defined as +1. Parentheses surrounding the HA tag in the PIDD1 reading frame represent the fact that the HA epitope sequence is encoded in the altPIDD1 reading frame, and is, therefore, undetected if expressed from the ATG codon at bp 1 of the PIDD1 coding sequence. PIDD1$^{\varnothing(HA)Flag}$ represents a monocistronic construct with the ATG translation initiation site for altPIDD1 deleted; the dashed box for altPIDD1 indicates that altPIDD1 cannot be expressed from this construct.

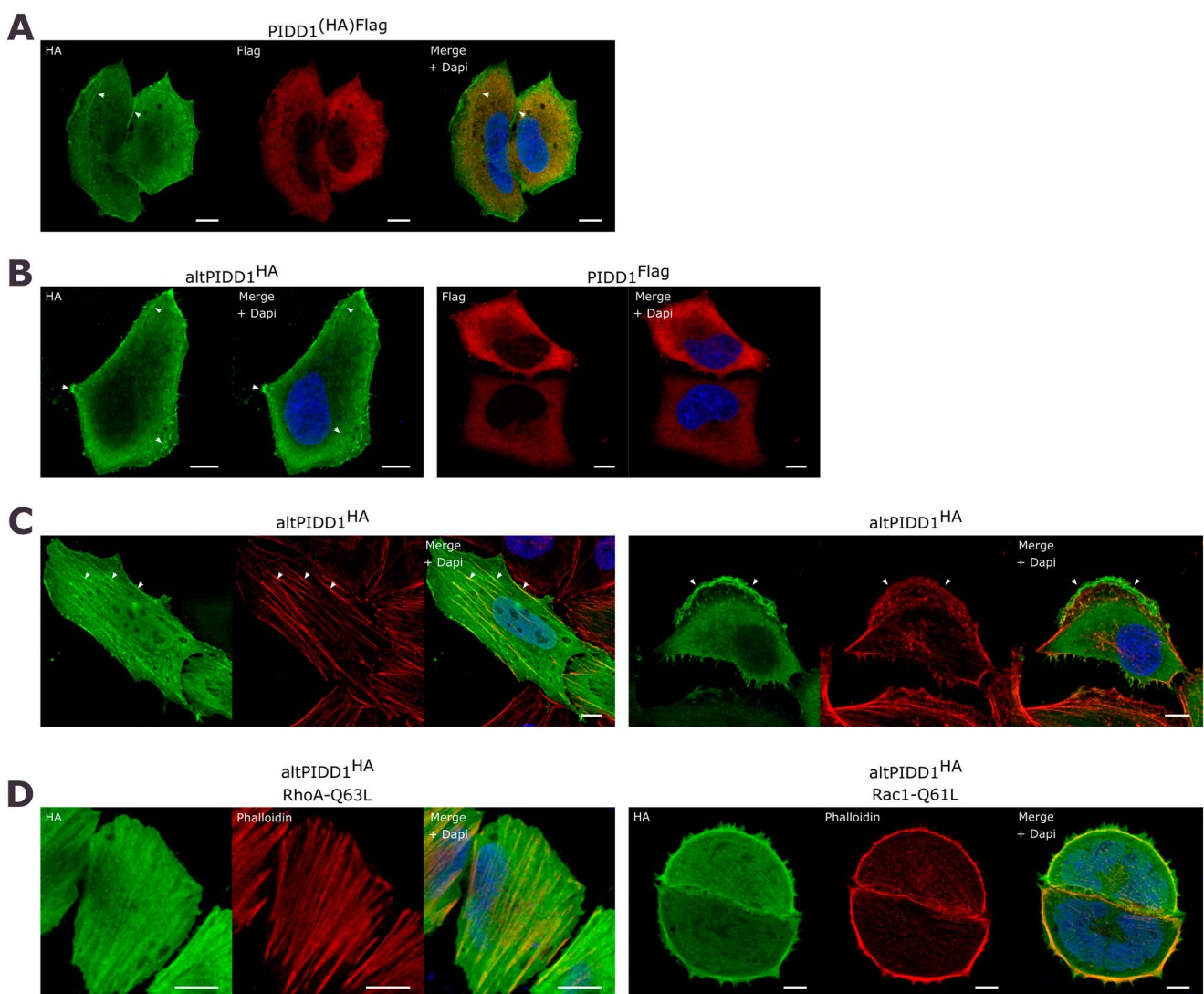

**Figure 3. AltPIDD1 localizes in actin-rich structures.**
**(A)** Images by confocal microscopy of altPIDD1 (HA tag, green) and PIDD1 (Flag tag, red) in HeLa cells transfected with PIDD1$^{(HA)Flag}$. Identically stained mock-transfected cells did not display any signal, highlighting the specificity of the observed signals. The white scale bar corresponds to 10 $\mu$m. Representative images of n = 3. Arrows indicate accumulation of altPIDD1 in cytoplasmic and membrane filamentous structures. **(B)** Images by confocal microscopy of altPIDD1 (HA tag, green) and PIDD1 (flag tag, red) in HeLa cells transfected with altPIDD1$^{HA}$ and PIDD1$^{FLAG}$, respectively. The white scale bar corresponds to 10 $\mu$m. Representative images of n = 3. Arrows indicate accumulation of altPIDD1 in cytoplasmic and membrane structures. **(C)** Images by confocal microscopy of altPIDD1 (HA tag, green) in cells labeled with phalloidin to show actin structures in migrating HeLa cells. Migration was induced by scratching a confluent cell layer 24 h before fixation. Left panel, cell with high levels of stress fibers (arrows). Right panel, cell with a large lamellipodium (arrows). The white scale bar corresponds to 10 $\mu$m. Representative images of n = 3. **(D)** Images by confocal microscopy of altPIDD1 (HA tag, green) and actin (phalloidin, red) in HeLa cells co-transfected with either RhoA-Q63L or Rac1-Q61L cells, as indicated, to induce the formation of stress fibers and lamellipodia, respectively. The white scale bar corresponds to 10 $\mu$m. Representative images of n = 3.

Tet$^R$PIDD1$^{(HA)HA}$ represents a dual-coding construct with both PIDD1 and altPIDD1 modified with an HA tag; expression of this construct in under the control of the tetracycline repressor. **(B)** PIDD1 auto-processing at residues S446 and S588 generate two C-terminal fragments. **(C)** Co-expression of both PIDD1 and altPIDD1 proteins from transfection of the PIDD1$^{(HA)Flag}$ construct in HeLa cells, by Western blot. Single-coding constructs altPIDD1$^{HA}$ and PIDD1$^{Flag}$ were used as positive controls and pcDNA was used as a transfection control. Actin was used as a loading control. Representative immunoblot from $n$ = 3. **(D)** Expression of altPIDD1 and PIDD1 from transfection of the PIDD1$^{(HA)Flag}$ and PIDD1$^{\emptyset(HA)Flag}$ constructs in HeLa cells, by Western blot. The translation initiation codon of altPIDD1 is deleted in the PIDD1$^{\emptyset(HA)Flag}$ construct. Actin was used as a loading control. Representative immunoblot from $n$ = 3. **(E)** Co-expression of altPIDD1 and PIDD1 from transfection of the Tet$^R$PIDD1$^{(HA)HA}$ construct in HeLa cells detected by immunoblotting. Expression was induced with doxycycline at the indicated concentration (ng/ml). Single-coding constructs altPIDD1$^{HA}$ and PIDD1$^{HA}$ are used as positive controls. GAPDH was used as a loading control. Representative Western blot from $n$ = 3. **(F)** Western blot analysis of PIDD1 and its processed fragments, and altPIDD1 in cells transfected with PIDD1$^{(HA)Flag}$ and treated with cycloheximide. Vinculin was employed as a control for ensuring equal loading, although c-myc was used as a control for a protein that undergoes degradation during cycloheximide treatment. **(G)** The endogenous nucleotide sequence surrounding the ATG initiation codon of altPIDD1 is displayed on the left, along with the sequence optimized by mutagenesis to correspond to an optimal Kozak motif (red nucleotides). Nucleotides in bold correspond to positions that are highly important for strong translation initiation. Left, expression of altPIDD1 and PIDD1 from transfection of the PIDD1$^{(HA)Flag}$ and PIDD1(Kozak)$^{(HA)Flag}$ constructs in HeLa, HEK293, and U2OS cells by immunoblotting. GAPDH was used as a loading control. Representative immunoblots from $n$ = 3.

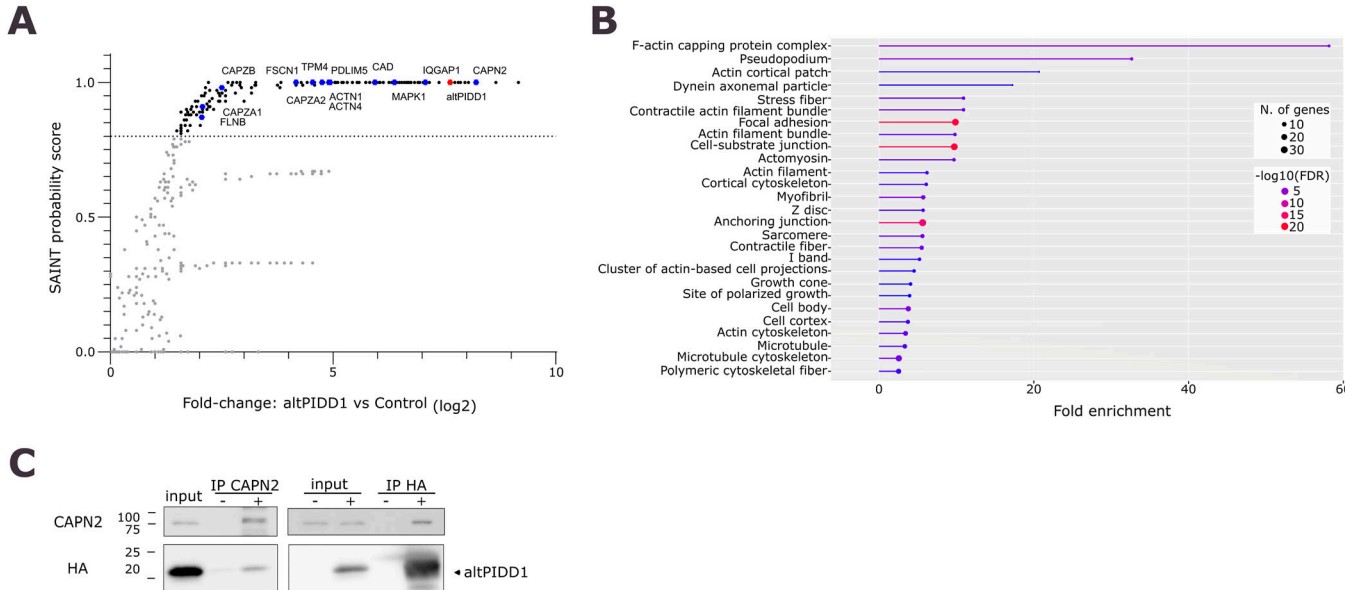

**Figure 4. AltPIDD1 interactome.**
**(A)** Scatter plot of the proteins identified by AP-MS indicating their enrichment (fold change over control) and their SAINT probability score. Proteins above the 0.8 threshold (gray line) are indicated in black, others in gray. AltPIDD1 is indicated in red (bait), and preys known to regulate the cytoskeleton are indicated in blue. **(B)** A gene ontology (cellular component) enrichment analysis was carried out on the proteins in the interactome of altPIDD1 using the ShinyGO v0.741 tool. The degree of enrichment for each ontology is indicated by the length of the bars on the x-axis. The number of enriched genes is indicated by the size of the dot at the end of the bar. The $-\log_2$ false discovery rate is indicated by the color of each bar and dot. Only GO terms related to parameters were set at a 0.05 FDR cutoff with no redundancy in GO terms. **(C)** Validation of the interaction between AltPIDD1 and calpain-2 (CAPN2) in HEK293a cells expressing altPIDD1[HA] by co-immunoprecipitation of endogenous CAPN2 (left) or co-immunoprecipitation of altPIDD1[HA] (right). Controls (−): immunoprecipitation performed without anti-CAPN2 antibodies (left), and immunoprecipitation performed on mock-transfected cells (right). Representative of n = 3.

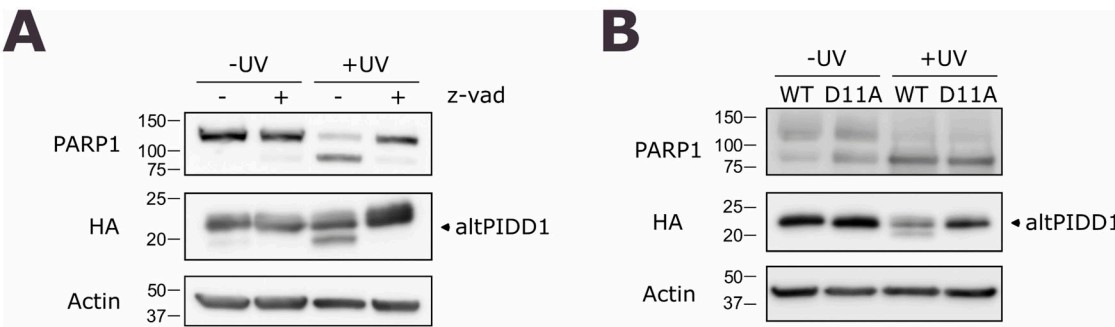

**Figure 5. AltPIDD1 is cleaved during apoptosis.**
**(A)** HeLa cells expressing altPIDD1[HA] were left untreated or treated with UV (254 nm wavelength, 130 J/m$^2$), in the absence or in the presence of and Z-VAD-fmk (10 $\mu M$), as indicated. Proteins were analyzed by immunoblotting using the indicated antibodies. **(B)** HeLa cells expressing WT or D11A altPIDD1[HA] were left untreated or treated with UV. Proteins were analyzed by immunoblotting using the indicated antibodies. Representative of n = 3.

contrast, control beads with no CAPN2 antibodies did not copurify altPIDD1[HA]. In a reciprocal experiment, immunoprecipitation of altPIDD1[HA] with anti-HA antibodies pulled down a significant fraction of endogenous CAPN2 (Fig 4C, right panel).

**AltPIDD1 contains a functional cleavage site for caspases 3 and 7**

Given that altPIDD1 is a likely intrinsically disordered protein and that these regions are enriched in short linear motifs (Holehouse & Kragelund, 2024), the sequence was analyzed using the Eukaryotic Linear Motif resource, which is a predictor of such motifs (Kumar

et al, 2020). A cleavage site for caspases 3 and 7 with the sequence S$^8$VGD*G is predicted at position 8–12. To test for the functionality of this site, cells expressing altPIDD1[HA] were irradiated with UV to induce caspase-mediated apoptosis. We used the cleavage of PARP1 as a reliable hallmark of apoptosis, which is cleaved by caspases 3 and 7 (Tewari et al, 1995; Boucher et al, 2012). As shown in Fig 5A, UV irradiation resulted in the significant cleavage of altPIDD1; the cleavage was prevented by the broad-spectrum caspase inhibitor z-VAD-fmk. To confirm the cleavage of altPIDD1 at D11, we generated the mutant D11A. This mutant was completely resistant to caspase cleavage after UV treatment (Fig 5B).

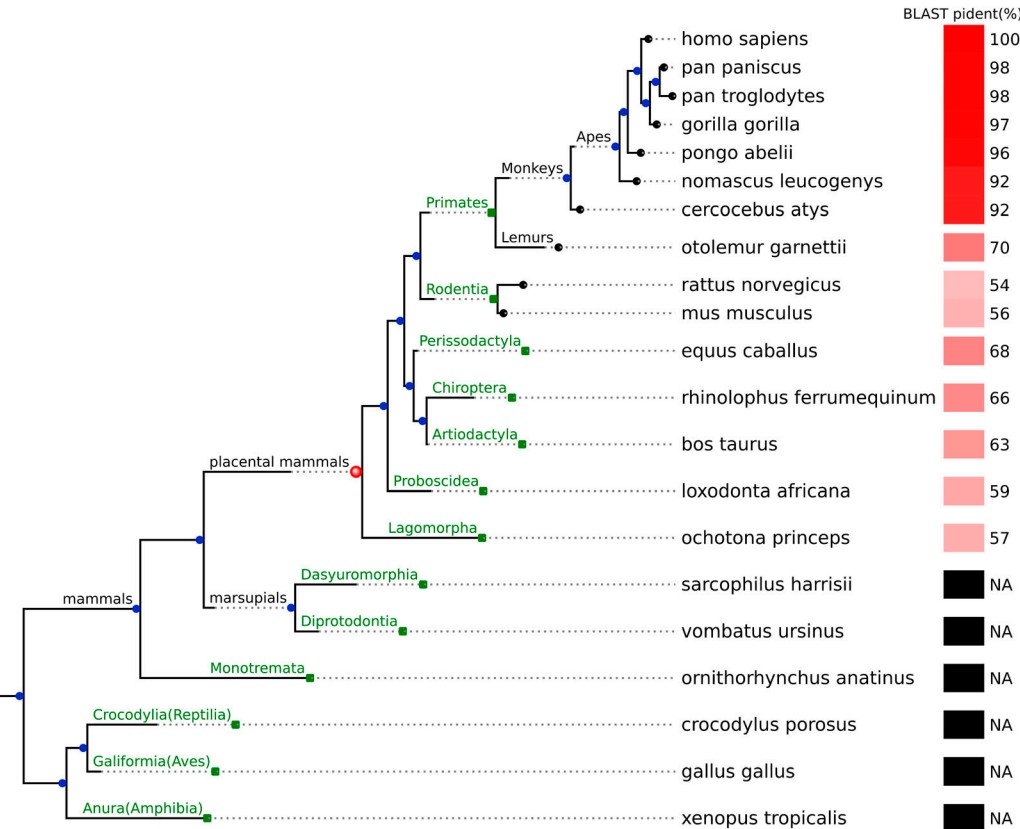

**Figure 6. Emergence of altPIDD1 in placental mammals.**
Gene tree ENSGT00940000161780 from Ensembl Compara, illustrating the evolutionary relationships among the one-to-one orthologs of human PIDD1 across various species. The leaves are represented by green squares, each corresponding to a taxonomic order, with one randomly selected species highlighted for each order. Details on specific species within the Primates and Rodentia orders are provided in subtrees with black leaves to highlight their importance in the study. Branch lengths represent the extent of evolutionary changes between the parent and the child nodes, and all internal nodes represent speciation events. Each species is associated with a percentage of identity (pident) derived from aligning altPIDD1 with its best matching protein in the ortholog of PIDD1 in this species using BLASTp. The pident values are visualized in a heatmap. A value of NA, depicted in black, indicates the absence of mapping, although a value of 100, depicted in red, corresponds to a perfect match between altPIDD1 and its best matching ORF. The pident heatmap reflects the conservation of altPIDD1 within the PIDD1 gene tree, with a particularly high conservation observed in mammals, especially placental mammals, but not in marsupials. Conservation is symbolized by a red sphere at the root of placental mammals. The highest pident is observed for monkeys and apes.

## AltPIDD1 has emerged in placental mammals

The amino acid sequence conservation of altPIDD1 within the genes of the *PIDD1* gene tree was assessed (Fig 6). We selected all one-to-one ortholog genes of PIDD1 from the NCBI orthologs database, spanning 465 species from 13 orders including primates to amphibians. For each ortholog of PIDD1, we considered all ORFs contained in the reference transcript, and selected the best matching ORF to altPIDD1 using MACSE alignment (Ranwez et al, 2018). For each pairwise alignment of altPIDD1 and its best matching protein in an ortholog gene, we report the percentage of identity (pident) derived from a NCBI protein BLAST alignment. It was found that altPIDD1 is conserved in placental mammals, with the highest level of conservation observed among primates. The absence of conservation in the outgroups of the placental mammals indicates that altPIDD1 emerged after PIDD1, specifically after the divergence of the placental mammals from the outgroup. Therefore, altPIDD1 does not belong to the category of noncanonical proteins translated from unannotated ORFs which are predominantly

evolutionarily young proteins that have emerged in primates (Sandmann et al, 2023).

## Discussion

The translation of noncanonical proteins from unannotated ORFs is receiving increasing attention (Ingolia et al, 2019; Hassel et al, 2023; Mohsen et al, 2023; Valdivia-Francia & Sendoel, 2024). Not only does this translation appear to be pervasive, involving several thousands of ORFs from yeast to humans, but it also implies that some mRNAs are multicoding, a property not usually associated with eukaryotic mRNAs. One of the issues in this field is to determine whether the translation of these proteins is due to simple biological noise. Here, we show that *PIDD1* contains a translated altORF that overlaps with the reference ORF in a different reading frame. This second protein, altPIDD1, is the primary translational product of this protein-coding gene.

A significant number of ribo-seq and proteomics data are available and can be reanalyzed with customized annotations and

protein sequence databases to detect alternative ORFs and their corresponding proteins. This proteogenomics approach is necessary for the detection of otherwise unannotated ORFs and proteins. We observed greater translation evidence for altPIDD1 than for PIDD1 using three resources: OpenProt, PepQuery, and GWIPZ. This prompted us to determine the ratio between both proteins. Using a transfected inducible system expressing the dual-coding transcript with the endogenous 5′UTR, we found that altPIDD1 was preferentially expressed at low induction levels. In non-transfected HEK293 cells, this ratio was estimated at 40 to 1. This observation is significant as it demonstrates that the principal translational product of *PIDD1* is not the reference protein, but altPIDD1. Both altPIDD1 and the processed PIDD1-CC form of PIDD1 are particularly stable, and the stoichiometry between altPIDD1 and PIDD1 is unlikely to result from a difference in protein stability. Interestingly, the translation of altPIDD1 is also reported in a manuscript currently under review (Deutsch et al, 2024 *Preprint*).

In our experiments, the co-expression of PIDD1 and altPIDD1 in transfected cells expressing a dual-coding transcript with the endogenous 5′UTR confirms that ribosomes can translate the two overlapping ORFs. Mechanistically, this means that ribosomes can recognize either one initiation codon or the other. Our data suggest that the translation of altPIDD1 is constitutive and that translation of PIDD1 is dependent on leaky scanning. This control of PIDD1 translation by altPIDD1 is supported by the observation that deletion of the altPIDD1 initiation codon resulted in higher levels of PIDD1 expression. Overall, the observed high ratio between the number of molecules of altPIDD1 and PIDD1 in the cell is likely mostly due to the preferential utilization of the altPIDD1 initiation codon.

Given that the expression of PIDD1 is dependent on leaky scanning through altPIDD1 initiation site, it is possible that the main function of altPIDD1 lies in its dampening of PIDD1 translation, and not in the produced protein itself. The *PIDD1* gene is a transcriptional target of p53 and E2F (Lin et al, 2000; Sladky et al, 2020), and the altPIDD1 ORF could act as a regulator at the translational level. However, it is unlikely that the role of altPIDD1 is exclusively related to the translation of PIDD1. Indeed, it is a stable protein that partially localizes and accumulates in actin-rich cytoskeletal structures associated with cell adhesion and cell motility. This was confirmed upon artificial formation of such structures by scratching a cell monolayer or by expressing two constitutively active GTPases. The functional relation between altPIDD1 and actin-rich cytoskeletal structures is supported by the enrichment of proteins involved in the regulation (e.g., calpain-2 (CAPN2/CAPNS1), MAPK1) or structural organization (e.g., ACTN1, ACTN4, FSCN1, FLNB) of these structures in the interactome of altPIDD1. The interaction between altPIDD1 and calpain-2 is of interest for several reasons. First, this interaction is specific since the other ubiquitous calpain, calpain-1, is not present in the interactome. Second, MAPK1, which is also highly enriched in the interactome, can directly activate calpain-2 by phosphorylating Ser50 (Glading et al, 2004; Zadran et al, 2010), a site that is not present in calpain-1. Finally, by cleaving adhesion complex components and migration-related proteins, calpain-2 increases cell motility (Franco & Huttenlocher, 2005). It can be hypothesized that altPIDD1 may be part of a platform for the

regulation of actin-rich cytoskeletal structures involved in cell migration. However, altPIDD1 interacts also with other cellular proteins, and it is premature to suggest a specific role in cell mobility. To determine its function, it will be necessary to eliminate its expression in cells. This will require a genome editing strategy specifically targeted at the altPIDD1 initiation codon to avoid any impact on the PIDD1 coding sequence.

PIDD1 or its primary interactors were not identified in the interaction network of altPIDD1. Nevertheless, because both proteins are encoded by the same gene and are obligatorily co-expressed, it would be interesting to determine if they are functionally related and act in the same biological pathway.

Following *MIEF1* (Delcourt et al, 2018), this is the second example of a multicoding gene with a stoichiometry in favor of the alternative protein. They exemplify the great importance to use proteogenomic strategies based on experimental ribo-seq and MS data with customized databases to improve gene annotation and recognize the contribution of multicoding genes to the proteome.

### Limitations of the study

In this article, the reanalysis of a variety of MS and ribo-seq data provided more evidence of expression for altPIDD1 than for PIDD1, a conclusion that was supported by absolute quantification of the two proteins in the widely used HEK293 cell line model. However, it is possible that the ratio between both proteins varies in different tissues and cell types, or in different conditions, and that PIDD1 expression outperforms that of altPIDD1. Although the primary objective of this study was to report the discovery of a novel protein and conduct preliminary characterization, the function of this protein remains to be demonstrated. Consequently, it is possible that it has no significant role in the cell. Although its expression level is considerably higher than that of PIDD1, and its interactome and appearance in placental mammals suggest an important role, it will be necessary to elucidate its molecular function. Finally, the localization of PIDD1 and altPIDD1 was determined in cells overexpressing both proteins. In cells overexpressing PIDD1, the protein localizes in the cytoplasm with no obvious accumulation in a specific cytoplasmic structure (Janssens et al, 2005; Tinel et al, 2007; Fig 3A and B). However, the localization of the endogenous protein shows both cytoplasmic and centrosomal localization (Burigotto et al, 2021; Evans et al, 2021). With regard to altPIDD1, the protein partially localized with actin. The identification of proteins from cytoskeletal structures in the interactome of altPIDD1 is in agreement with the observed localization in cells overexpressing the protein but this will have to be confirmed with endogenous altPIDD1.

# Materials and Methods

### Constructs

PIDD1 and altPIDD1 constructs (RefSeq NM_145886.3, Ensembl ENST00000347755.9) were obtained from Bio Basic Gene Synthesis service and cloned into pcDNA3.1- (Invitrogen) using Gibson

assembly (E26115; New England Biolabs). PIDD1 and altPIDD1 proteins were either untagged or C-terminally tagged with two Flag (DYKDDDDKDYKDDDDK) or HA (YPYDVPDYA), respectively. For the doxycycline-inducible Tet-On system, the PIDD1$^{(HA)HA}$ construction was cloned into a modified pgLAP5 plasmid in which the C-terminal S peptide, Prescission Protease site and EGFP were removed. Monocistronic PIDD1$^{\emptyset(HA)Flag}$, PIDD1$^{(HA)FLAG}$ Kozak + and alt-PIDD1(D11A)$^{HA}$ constructs were generated by site-directed mutagenesis of PIDD1$^{(HA)FLAG}$ and altPIDD1$^{HA}$ constructs, respectively. All constructs were confirmed by sequencing. pcDNA3-EGFP-RhoA-Q63L was a gift from Gary Bokoch (plasmid # 12968; http://n2t.net/addgene:12968; RRID:Addgene_12968; Addgene). pcDNA3-EGFP-Rac1-Q61L was a gift from Gary Bokoch (plasmid # 12981; http://n2t.net/addgene:12981; RRID:Addgene_12981; Addgene).

### Cell culture, transfection, and treatment

HEK293 and HeLa cells lines cultures tested routinely negative for mycoplasma contamination. Transfections were carried out with jetPRIME reagent (101000046; Polyplus) according to the manufacturer's instructions. Where indicated, doxycycline solubilized with DMSO or DMSO was added to cells 24 h post-transfection, followed by an incubation for 24 h before being harvested.

### Western blot analysis

After 24 h of transfection, cells in six-well plates were washed with PBS and lysed in mRIPA buffer (1% Triton, 1% NaDeoxycholate, 0.1% SDS, 1 mM EDTA, 50 mM Tris–HCl, pH 7.5) supplemented with protease inhibitor cocktail (11836170001; Roche). The lysate was thoroughly vortexed, sonicated, and centrifuged at 16,000$g$ for 10 min before quantification of proteins using the BCA protein assay reagent (23223; Pierce, 23224; Pierce). Depending on the experiment, 50–100 $\mu$g of protein was diluted in 4X SDS–PAGE loading dye. After electrophoresis, proteins were transferred to a PVDF membrane according to the manufacturer's instructions. Blocking was performed at RT with low agitation for 30 min in 5% milk/TBS-T (5% wt/vol powder milk, 10 mM Tris, pH 8.0, 150 mM NaCl, 1% Tween-20). Primary antibodies were diluted in the same buffer as follows: anti-Flag (F1804; Sigma-Aldrich) 1:4,000, anti-HA (26183; Thermo Fisher Scientific) 1:4,000, anti-actin (A5541; Sigma-Aldrich) 1:8,000, anti-GAPDH (5174; Cell Signaling) 1:10,000, anti-c-Myc/N-Myc (13987; Cell Signaling) 1:1,500, anti-vinculin (13901; Cell Signaling) 1:1,000. After 16 h at 4$^{o}$C with low agitation, membranes were washed three times with TBS-T before gentle shaking for 1 h at RT with secondary antibodies diluted as follows: m-IgGk BP-HRP (sc-516102; Santa Cruz Biotechnology) 1:8,000, anti-rabbit HRP (7074; Cell Signaling) 1:8,000. Membranes were then washed three times with TSB-T before revelation using ECL reagent (1705061; Bio-Rad) and high-resolution image acquisition using ImageQuant LAS 4000 (GE Healthcare Life Sciences).

### UV treatment

After 24 h of transfection, media was replaced with PBS for UV treatment. UV irradiation was performed with a dose of 130 J/m$^2$ (254 nm), measured with a UVX radiometer (Ultra-Violet Products). PBS was immediately replaced with fresh media after the treatment. Cells were harvested 24 h post-irradiation for immunoblotting.

### Immunofluorescence

Cells were grown on 12-mm glass coverslips for 24 h before transfection. 24 h after transfection, cells were washed with PBS and fixed for 20 min with 4% PFA and 4% sucrose. After washing, cells were permeabilized with 0.15% Triton X-100 for 5 min. Cells were washed with PBS and blocking was performed for 30 min with 10% normal goat serum (053-150; Wisent) diluted in PBS. Coverslips were incubated overnight in a humidified chamber at RT with the following antibodies diluted with 10% normal goat serum as follows: anti-Flag (F1804; Sigma-Aldrich) 1:1,000, anti-HA (26183; Thermo Fisher Scientific) 1:1,000. After three washing steps with blocking buffer, the following secondary antibodies were diluted as follows and added to the coverslips for 1 h: goat anti-rabbit, Alexa Fluor 488 (A11008; Invitrogen) 1:1,000, goat anti-mouse, Alexa Fluor 647 (A21236; Invitrogen) 1:1,000, goat anti-rabbit, Alexa Fluor 647 (A21245; Invitrogen) 1:1,000. Cells were then washed with PBS, incubated for 10 min with 0.25 $\mu$g/ml DAPI (10236276001; Sigma-Aldrich) and washed again before mounting with SlowFade (S36972; Invitrogen) on glass slides for observation. For phalloidin staining, phalloidin-iFluor 555 (176756; Abcam) was diluted 1:1,000 and added during the incubation with secondary antibodies.

### Confocal analysis

Slides were examined with a scanning confocal microscope (LSM700; Zeiss) coupled to an inverted microscope with a 63× oil immersion objective (Plan-APOCHROMAT). Specimens were laser-excited at 405 nm (DPSS 405 nm), 488 nm (DPSS 488 nm), 555 nm (DPSS 555 nm), and 639 nm (DPSS 639 nm). Pixel sizes vary from 0.07 × 0.07 $\mu$m to 0.12 × 0.12 $\mu$m according to image size. To avoid cross-talk, fluorescence was collected sequentially. For illustration, purposes images were pseudocolored according to their original fluorochromes, except for phalloidin-555 which was pseudocolored red for clarity in Fig 3, panel D, where alexa-647 tagged altPIDD1-HA was pseudocolored green for consistency with previous panels. Colocalization analysis was performed directly with ZEN 2.6 blue edition (Zeiss). Images were cropped and assembled using Inkscape.

### Endogenous detection of altPIDD1 using parallel reaction monitoring (PRM)

Exponentially growing HeLa WT cells were lysed in 8 M urea (in 10 mM HEPES). 100 $\mu$g of the cell extracts were reduced in 10 mM DTT, boiled, and alkylated in 7.5 mM 2-Chloroacetamide for 30 min in the dark. The urea concentration in the lysate was reduced to 2 M with the addition of 50 mM NH$_4$HCO$_3$, and the samples were subjected to overnight trypsin digestion (Trypsin Gold, MS Grade, Promega Corporation). After digestion, the extracted peptides were desalted using zip-tips, dried by speedvac and resuspended in 1% formic acid. The PRM method was first determined by identification of peptides that were detectable in overexpression of altPIDD1 in a shotgun-based approach. Peptide unicity was checked using

neXtProt peptide uniqueness checker (MacLean et al, 2010). MS/MS spectra were then manually inspected using Skyline (Pino et al, 2020) and peptides with highest MS intensities, absence of mis-cleavage and high identification scores were selected for the final PRM method. For altPIDD1, one peptide was identified as such: ATPGHTGCLSPGCPDQPAR. PRM analysis was then performed on a HEK293a whole cell lysate and injection of a pure ATPGHTGCLSPGCPDQPAR synthetic peptide was used as a control. Peptides were analyzed on an OrbiTrap QExactive (Thermo Fisher Scientific) as previously described (Dubois et al, 2020). The Universal Spectrum Explorer tool (Schmidt et al, 2021) was used to create the mirror panel shown in Fig 1F.

### Peptide fractionation, multiple reaction monitoring MS, and relative quantification using stable isotopes

Cell lysis and processing for MS was performed as described above. Before peptide fractionation, cell lysates were spiked with a known concentration of synthetic stable isotope-labeled peptides (Gerber et al, 2003) (AQUA) obtained from Life Technologies: ATPGHTG[C]LSPG[C]PDQPA(R) for altPIDD1, LQSLPASLAGL(R) and VNLIALQ(R) for PIDD1. Cysteines in the heavy peptides are carbamidomethylated and C-terminal arginines are +10 Da (U-13C6, 15N4). Fractionation was performed with 2 mg Strata-X solid phase extraction cartridges (8M-S100-4GA; Phenomenex). Columns were conditioned with 100% Acetonitrile (ACN) before washing with 5% $NH_4OH$, loading 75 $\mu g$ of peptides and washing again. Peptides were eluted into six fractions with increasing ACN concentrations in 5% $NH_4OH$. Peptides were then dried and resuspended in 3% DMSO, 0.2% formic acid before loading. Acquisition was performed with a ABSciex TripleTOF 6600 (ABSciex) equipped with an electrospray interface with a 25 $\mu m$ iD capillary and coupled to an Eksigent $\mu$UHPLC (Eksigent). Analyst TF 1.8 software was used to control the instrument and for data processing and acquisition. Acquisition was performed in Production Ion Scan (PIS). The source voltage was set to 5.35 kV and maintained at 325°C, curtain gas was set at 50 psi, gas one at 40 psi and gas two at 35 psi. Separation was performed on a reversed phase Kinetex XB column 0.3 $\mu m$ i.d., 2.6 $\mu m$ particles, 150 mm long (Phenomenex) which was maintained at 60°C. Samples were injected by loop overfilling into a 5 $\mu l$ loop. For the 15 min LC gradient, the mobile phase consisted of the following solvent A (0.2% vol/vol formic acid and 3% DMSO vol/vol in water) and solvent B (0.2% vol/vol formic acid and 3% DMSO in EtOH) at a flow rate of 3 $\mu l$/min. Relative quantification was performed by comparing the areas of the endogenous peptides to the stable isotopes in the fraction in which the peptides had the highest intensity for each replicate.

### Affinity purification–mass spectrometry (AP–MS) and MS analysis

AP–MS was performed as described previously (Leblanc et al, 2023) with anti-Flag magnetic beads (M8823; Millipore Sigma). MS analysis was performed as described previously (Brunet et al, 2021) with the UniProt Knowledgebase (*Homo sapiens*, Swiss-Prot, released in October 2020) with the addition of the sequence of altPIDD1 (IP_191523). The SAINT algorithm was used to score protein interactions, using mock-transfected cells and the HeLa Flag beads in the CRAPome 2.0 database (Mellacheruvu et al, 2013) as controls (#CC130, CC135, CC136, CC137, CC139, CC140).

Co-IP experiments to validate the interaction between altPIDD1 and CAPN2 were performed as described previously (Leblanc et al, 2023) with anti-Flag magnetic beads (M8823; Millipore Sigma) for altPIDD1-HA and A/G magnetic beads (B23201; Cedarlane) with or without anti-CAPN2 antibodies (ab39165; Abcam) for CAPN2. Western blotting was performed as described above.

### Peptide-centric analysis of proteomics data sets

AltPIDD1 protein sequence was submitted to PepQuery 2.0 with the following parameters: task type = novel peptide/protein; target event = protein sequence; reference database = swissprot human 20220527; scoring algorithm = hyperscore; fast searching mode disabled. PIDD1 protein sequence was submitted with the same parameters except the following: task type = known protein. All 48 available MS/MS datasets were queried directly from the PepQuery website.

### Ribo-seq data

Global aggregate reads for initiating ribosomes and elongating ribosomes footprints across all available studies were downloaded from the Gwips portal (https://gwips.ucc.ie/).

### Conservation analyses

For altPIDD1 protein conservation analysis, all available PIDD1 one-to-one ortholog genes were obtained from NCBI orthologs (one ortholog gene per species). We performed an in silico three-frame translation and retrieved all ORFs contained in the reference transcript of each ortholog gene. Next, the best matching ORF per ortholog gene was selected by pairwise alignment with altPIDD1 sequence using MACSE (Ranwez et al, 2018). Percentages of identity (pident) of the pairwise alignments of altPIDD1 with its best matching protein in ortholog genes were obtained using the alignment feature of NCBI protein BLAST (https://blast.ncbi.nlm.nih.gov/Blast.cgi). Protein sequences from species in which BLASTp was unable to align with human altPIDD1 were considered not conserved. The PIDD1 evolutionary tree, in which altPIDD1 conservation data were appended, was obtained from the Ensembl Compara database V11 (Ensembl identifier ENSGT00940000161780).

### Cellular compartment enrichment analysis

Proteins identified in the interactome of altPIDD1 were screened for cellular compartment ontology enrichment using the ShinyGO v0.741 tool (Ge et al, 2020). *P*-value cutoff (FDR) was set at 0.05.

# Supplementary Information

## Acknowledgements

This research was supported by a CIHR grant and by a Canada Research Chair in Functional Proteomics and Discovery of Novel Proteins to X Roucou. We thank the Allumiqs proteomics platform and the proteomics and microscopy platforms at Université de Sherbrooke for their collaboration. We thank Thomas Gardrat in the laboratory of Dr Conconi at Université de Sherbrooke for operating the UVX radiometer. We thank The laboratories of M Labrie and P Lavigne for sharing the antibodies against vinculin and c-myc, respectively.

### Author Contributions

F Comtois: conceptualization, data curation, formal analysis, investigation, visualization, methodology, and writing—original draft.
J-F Jacques: formal analysis, supervision, validation, investigation, methodology, and writing—review and editing.
L Métayer: formal analysis, validation, and visualization.
WYDD Ouedraogo: formal analysis, investigation, visualization, methodology, and writing—review and editing.
A Ouangraoua: supervision, methodology, and writing—review and editing.
J-B Denault: resources, methodology, and writing—review and editing.
X Roucou: conceptualization, resources, formal analysis, supervision, funding acquisition, investigation, and writing—original draft and project administration.

### Conflict of Interest Statement

The authors declare that they have no conflict of interest.

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
