## [Reviewer comments · Life Science Alliance]

Life Science Alliance

Non-canonical altPIDD1 protein: unveiling the true major translational output of the PIDD1 gene

Frédéric Comtois, Jean-François Jacques, Lenna Métayer, Wend Yam Ouedraogo, Aïda Ouangraoua, Jean-Bernard Denault, and Xavier Roucou

DOI: <https://doi.org/10.26508/lsa.202402910>

Corresponding author(s): *Xavier Roucou, Université de Sherbrooke*

Review Timeline:

Submission Date:	2024-06-26
Editorial Decision:	2024-08-09
Revision Received:	2024-11-01
Editorial Decision:	2024-11-04
Revision Received:	2024-11-04
Accepted:	2024-11-05

Scientific Editor: *Eric Sawey, PhD*

Transaction Report:

August 9, 2024

Re: Life Science Alliance manuscript #LSA-2024-02910

Dr. Xavier Roucou
Université de Sherbrooke
Biochemistry
3201 Jean Mignault
3001 12eme avenue nord
Sherbrooke, QC J1E4K8
Canada

Dear Dr. Roucou,

Thank you for submitting your manuscript entitled "Non-canonical altPIDD1 protein: unveiling the true major translational output of the PIDD1 gene" to Life Science Alliance. The manuscript was assessed by expert reviewers, whose comments are appended to this letter. We invite you to submit a revised manuscript addressing the Reviewer comments.

Thank you for this interesting contribution to Life Science Alliance. We are looking forward to receiving your revised manuscript.

Sincerely,

B. MANUSCRIPT ORGANIZATION AND FORMATTING:

Reviewer #1 (Comments to the Authors (Required)):

altPIDD1: In the work submitted here, the authors show by exploring proteogenomic datasets that a second more abundant polypeptide is translated from the dominant PIDD1 mRNA transcript that results in the generation of a 171aa protein. This protein appears significantly more abundant than canonical PIDD1 and does not constitute an isoform, as it is encoded in an alternative reading frame. Corresponding peptides are found in multiple proteomics datasets from different tissues, phylogenetic analysis suggests that this second protein is found in placental mammals.

The authors provide evidence that this first ORF dampens translation of the second, canonical PIDD1 transcript, by mutagenesis of the suboptimal KOZAK sequence in PIDD1. Yet, the authors also conclude that this may not be its sole function. Overexpression, IF and MS analyses and suggests that the altPIDD1 protein associates with cytoskeletal components and is found to enrich in stress fibers and filopodia.

Comments:

- 1) be consistent in text and figures to call it PIDD1; PIDD1-C; PIDD1_CC
- 2) Table S3 contains no data?!
- 3) The CHX experiment needs a positive control, blotting for a labile protein, e.g. MCL1
- 4) Please acknowledge centrosomal localization of PIDD1 and discuss your findings in overexpression
- 5) Related to the above, it could be mentioned that PIDD1 mutant mice, Manzl et al JBC 2009 likely also lack expression of altPIDD1, yet phenotypes are comparable between PIDD1, RAIDD and C2 mutant mice, at least in liver ploidy (Sladky, Dev Cell, 2020)
- 6) I suggest to fuse Figs 4-6, there is no reason to spread the data out that much
- 7) The text should be checked for wrong call outs to figures and wrong concentrations. For instance: page 10, concentrations are mentioned as (0.05) mg/ml, but in figure legend 2 as ng/ml; on page 13 Figure 6 is referred to as Fig. 5
- 8) It should be stated how many cells were looked at for the immunofluorescence and not only how often the experiment was performed, maybe some provide some quantification and assign categories X/Y cells show enrichment of altPIDD1 in filopodia

Reviewer #2 (Comments to the Authors (Required)):

In this study, Comtois et al provide evidence that a 171 amino acid alternative protein, alt-PIDD1, is the major translational output of the PIDD1 gene, not the well-characterized reference ORF. The quantitative expression analysis of endogenously expressed altPIDD1 is rigorous, and further supports the authors' previous conceptual advance (in the context of MIEF1) that altORFs can supersede refORFs in expression. They provide some molecular characterization of the interactions and localization of the overexpressed, epitope-tagged altORF. While the molecular and cellular assays rely exclusively on overexpression, an approach with clear limitations, the authors draw reasonable conclusions that are in line with the assays performed. Overall, the study extends the conceptual framework of altORFs to a new gene and provides a useful resource for the field. I have only minor comments.

1. The ratio of altPIDD1 to PIDD1 was measured only in HEK cells; while this was acknowledged as a limitation, it would be useful to understand if this is true in additional cell lines in order to draw the conclusion that altPIDD1 is indeed the major gene

product. It is not necessary to repeat the AQUA analysis if arduous; given that their relative levels appear to be predominantly translationally controlled, perhaps a simple reanalysis of RIBO-seq data in a few different cell types could provide the necessary insight, or transfection of their dual expression reporter in a different cell type to determine whether leaky scanning consistently regulates their relative levels in more than one system. These authors could easily complete such an analysis or experiment in days.

2. The colocalization of altPIDD1 with the cytoskeleton is low under almost all conditions presented. This may be due to the altPIDD1 overexpression system utilized. One approach that could definitively demonstrate colocalization is the proximity ligation assay; another would be endogenous genomic locus tagging with CRISPR-directed homology repair - again, it is not necessary to perform such an experiment, but it would be merited to mention these as limitations of the study.

3. Reliance on overexpression for molecular and cellular studies should be mentioned as a limitation of the study.

4. The authors should clearly state that the protein has no regions of predicted secondary structure; describing the "ribbon" structure is a bit confusing. It is not necessary to describe the protein as "disordered," which is also a poorly defined term and may not be true in complex with binding partners.

5. Several minor errors are present in the figures:

a. Check spelling of "stoichiometry" on y axis in Figure 1G.

b. Figure 4A, legend indicates that altPIDD1 will be indicated in red, but it is not.

Reviewer #3 (Comments to the Authors (Required)):

In this study, Comtois et al. document that a second 171-amino acid protein (altPIDD1) is translated from an overlapping alternate reading frame from the mRNA encoding the 910-amino acid PIDD1 protein. The authors also show that the levels of the altPIDD1 protein are significantly higher than the levels of the PIDD1 in the cells examined. While translation of altPIDD1 impacts translation of PIDD1, the altprotein likely has a separate function as it is found localized with actin-rich structures in the cell and co-immunoprecipitates with the CAPN2 subunit of calpain-2.

The results are convincing and clearly presented. I only have relatively minor comments:

1. In Figures 2, 4 and 5: it would be useful to have all protein bands labeled on the right (while keeping the tag for detection labeled on the left).

2. More of the blots should be shown in Figure 4C (as in Figure 2).

3. The authors should consider the possibility that the ratio of altPIDD1 to PIDD1 might change in some cell types or under some conditions.

We would like to express our gratitude to the three reviewers for their insightful and constructive comments, which have enabled us to enhance the manuscript and clarify several points. Please find our detailed responses below. The revised text in the manuscript is in red.

Reviewer #1

Comment #1: Be consistent in text and figures to call it PIDD1; PIDD1-C; PIDD1_CC

Response: We made the appropriate modifications to be consistent and use the terms PIDD1, PIDD1-C and PIDD1-CC in the text and in the figures.

Comment #2: Table S3 contains no data

Response: Table S3 is an Excel file with two sheets: sheet#1 contains a brief legend and sheet #2 contains the data. We do not understand why the reviewer had no access to the data and we are sorry about that.

Comment #3: The CHX experiment needs a positive control, blotting for a labile protein, e.g. MCL1.

Response: We agree with the reviewer that a positive control was needed. We have blotted the same samples with anti-MYC antibodies. The results confirm that the cycloheximide treatment worked since we observe a large reduction in the levels of MYC. We have also probed the blot with anti-vinculin antibodies to check for equal loading. We have modified figure 2F and the text accordingly.

Comment #4: Please acknowledge centrosomal localization of PIDD1 and discuss your findings in overexpression.

Response: As suggested by the reviewer, we acknowledge centrosomal localization of PIDD1 in the discussion. We have added the following text and corresponding references, in response to this comment and to comment #2 from reviewer #2 (see below): *“Finally, the localization of PIDD1 and altPIDD1 was determined in cells overexpressing both proteins. In cells overexpressing PIDD1, the protein localizes in the cytoplasm with no obvious accumulation in a specific cytoplasmic structure (Janssens et al, 2005; Tinel et al, 2007; Fig. 3A, B). However, the localization of the endogenous protein shows both cytoplasmic and centrosomal localization (Burigotto et al, 2021; Evans et al, 2021). With regards to altPIDD1, the protein partially localized with actin. The identification of proteins from cytoskeletal structures in the interactome of altPIDD1 is in agreement with the observed localization in cells overexpressing the protein but this will have to be confirmed with endogenous altPIDD1”.*

Comment #5: Related to the above, it could be mentioned that PIDD1 mutant mice, Manzl et al JBC 2009 likely also lack expression of altPIDD1, yet phenotypes are comparable between PIDD1, RAIDD and C2 mutant mice, at least in liver ploidy (Sladky, Dev Cell, 2020).

Response: In mice, the altPIDD1 coding sequence is entirely nested within exon 2. In Manzl's manuscript, the KO strategy deleted exons 3 to 15 and left exons 1, 2 and 16 intact. It is therefore

likely that PIDD1 KO mice retain altPIDD1 expression. However, this hypothesis remains to be confirmed. In the absence of such confirmation, it seems too speculative to discuss in the article the point raised by the reviewer.

Comment #6: I suggest to fuse Figs 4-6, there is no reason to spread the data out that much

We have been unable to identify a logical way for integrating the findings of these three figures into a single, comprehensive representation. This is because they address distinct aspects, including the interactome, the caspase site, and conservation. Since the other reviewers did not have the same suggestion, we have left the figures unchanged.

Comment #7: The text should be checked for wrong call outs to figures and wrong concentrations. For instance: page 10, concentrations are mentioned as (0.05) mg/ml, but in figure legend 2 as ng/ml; on page 13 Figure 6 is referred to as Fig. 5

Response: We thank the reviewer for pointing out these discrepancies. We have carefully checked the text and made the appropriate corrections.

Comment #8: It should be stated how many cells were looked at for the immunofluorescence and not only how often the experiment was performed, maybe some provide some quantification and assign categories X/Y cells show enrichment of altPIDD1 in filopodia.

Response: The reviewer mentions filopodia but we did not specifically investigate these structures. However, we agree that we should state how many cells were looked at. We have added the following text in the paragraph related to figure 3: *In these experiments, a minimum of 100 cells were observed for each replicate and altPIDD1 always localized in actin structures labeled with phalloidin.*

Reviewer # 2

Comment #1: The ratio of altPIDD1 to PIDD1 was measured only in HEK cells; while this was acknowledged as a limitation, it would be useful to understand if this is true in additional cell lines in order to draw the conclusion that altPIDD1 is indeed the major gene product. It is not necessary to repeat the AQUA analysis if arduous; given that their relative levels appear to be predominantly translationally controlled, perhaps a simple reanalysis of RIBO-seq data in a few different cell types could provide the necessary insight, or transfection of their dual expression reporter in a different cell type to determine whether leaky scanning consistently regulates their relative levels in more than one system. These authors could easily complete such an analysis or experiment in days.

Response: As suggested by the reviewer, we have transfected the dual expression reporter in two other cell lines, HEK293 and U2OS. The results confirm leaky scanning, indicating similar level of regulation in three different cell lines. The text and Fig. 2G have been modified accordingly.

Comment #2: The colocalization of altPIDD1 with the cytoskeleton is low under almost all conditions presented. This may be due to the altPIDD1 overexpression system utilized. One approach that could definitively demonstrate colocalization is the proximity ligation assay; another would be endogenous genomic locus tagging with CRISPR-directed homology repair - again, it is not necessary to perform such an experiment, but it would be merited to mention these as limitations of the study.

Response: We agree with the reviewer and we have added the following sentence in the limitations, in response to this comment and to comment #4 from reviewer #1 (see above): We have added the following text and corresponding references, in response to this comment and to comment #2 from reviewer #2 (see below): *“Finally, the localization of PIDD1 and altPIDD1 was determined in cells overexpressing both proteins. In cells overexpressing PIDD1, the protein localizes in the cytoplasm with no obvious accumulation in a specific cytoplasmic structure (Janssens et al, 2005; Tinel et al, 2007; Fig. 3A, B). However, the localization of the endogenous protein shows both cytoplasmic and centrosomal localization (Burigotto et al, 2021; Evans et al, 2021). With regards to altPIDD1, the protein partially localized with actin. The identification of proteins from cytoskeletal structures in the interactome of altPIDD1 is in agreement with the observed localization in cells overexpressing the protein but this will have to be confirmed with endogenous altPIDD1”*.

Comment #3: Reliance on overexpression for molecular and cellular studies should be mentioned as a limitation of the study.

Response: We agree with the reviewer. This is now mentioned in response to reviewer #1 comment #4 and in response to the above comment. We’ve added the following lines: *“Finally, the localization of PIDD1 and altPIDD1 was determined in cells overexpressing both proteins. In cells overexpressing PIDD1, the protein localizes in the cytoplasm with no obvious accumulation in a specific cytoplasmic structure (Janssens et al, 2005; Tinel et al, 2007; Fig. 3A, B). However, the localization of the endogenous protein shows both cytoplasmic and centrosomal localization (Burigotto et al, 2021; Evans et al, 2021). With regards to altPIDD1, the protein partially localized with actin. The identification of proteins from cytoskeletal structures in the interactome of altPIDD1 is in agreement with the observed localization in cells overexpressing the protein but this will have to be confirmed with endogenous altPIDD1”*.

Comment #4: The authors should clearly state that the protein has no regions of predicted secondary structure; describing the "ribbon" structure is a bit confusing. It is not necessary to describe the protein as "disordered," which is also a poorly defined term and may not be true in complex with binding partners.

Response: As suggested by the reviewer, we now clearly state that the protein has no predicted secondary structures and we have removed the information about a ribbon-like structure. Regarding the disordered structure prediction, we think it is important to leave this information. Indeed, the biological importance of intrinsically disordered domains and proteins is clearly recognised and well demonstrated in the literature. We have modified the text accordingly.

Previous text: *The 3D structure predicted by AlphaFold2 indicates a predominantly ribbon-like structure (Fig. S1C). However, the confidence scores of these predictions are not particularly high, supporting the likely disordered nature of the protein.*

Revised text: *The 3D structure predicted by AlphaFold2 indicates no regions of secondary structures (Fig. S1C).*

Comment #5: Several minor errors are present in the figures:

a. Check spelling of "stoichiometry" on y axis in Figure 1G.

b. Figure 4A, legend indicates that altPIDD1 will be indicated in red, but it is not.

Response: we thank the reviewer for pointing out these errors. We have made the appropriate corrections.

Reviewer #3

Comment #1: In Figures 2, 4 and 5: it would be useful to have all protein bands labeled on the right (while keeping the tag for detection labeled on the left).

Response: We have made the appropriate changes.

Comment 2: More of the blots should be shown in Figure 4C (as in Figure 2).

Response: We have made the appropriate changes to match figure 2.

Comment 3: The authors should consider the possibility that the ratio of altPIDD1 to PIDD1 might change in some cell types or under some conditions.

Response: The reviewer is absolutely right and that is why we had already written in the Limitations of the study, at the end of the Discussion, that "*However, it is possible that the ratio between both proteins vary in different tissues and cell types, or in different conditions, and that PIDD1 expression outperforms that of altPIDD1*". We believe this sentence address the reviewer's comment.

November 4, 2024

RE: Life Science Alliance Manuscript #LSA-2024-02910R

Dr. Xavier Roucou
Université de Sherbrooke
Biochemistry
3201 Jean Mignault
3001 12eme avenue nord
Sherbrooke, QC J1E4K8
Canada

Dear Dr. Roucou,

Thank you for submitting your revised manuscript entitled "Non-canonical altPIDD1 protein: unveiling the true major translational output of the PIDD1 gene". We would be happy to publish your paper in Life Science Alliance pending final revisions necessary to meet our formatting guidelines.

- please be sure that the authorship listing and order is correct
- please add the Twitter handle of your host institute/organization as well as your own or/and one of the authors in our system

A. FINAL FILES:

B. MANUSCRIPT ORGANIZATION AND FORMATTING:

Sincerely,

November 5, 2024

RE: Life Science Alliance Manuscript #LSA-2024-02910RR

Dr. Xavier Roucou
Université de Sherbrooke
Biochemistry
3201 Jean Mignault
3001 12eme avenue nord
Sherbrooke, QC J1E4K8
Canada

Dear Dr. Roucou,

Thank you for submitting your Research Article entitled "Non-canonical altPIDD1 protein: unveiling the true major translational output of the PIDD1 gene". It is a pleasure to let you know that your manuscript is now accepted for publication in Life Science Alliance. Congratulations on this interesting work.

DISTRIBUTION OF MATERIALS:

Again, congratulations on a very nice paper. I hope you found the review process to be constructive and are pleased with how the manuscript was handled editorially. We look forward to future exciting submissions from your lab.

Sincerely,
